# Effect of Carcinomas on Autosomal Trait Screening: A Review Article

Husein Alhatim [1], Muhammad Nazrul Hakim Abdullah [1],*, Suhaili Abu Bakar [1] and Sayed Amin Amer [2],*

[1] Department of Biomedical Sciences, Faculty of Medicine and Health Sciences, Universiti Putra Malaysia, Serdang 43400, Selangor, Malaysia; hah1412@hotmail.com (H.A.); suhaili_ab@upm.edu.my (S.A.B.)

[2] Department of Forensic Sciences, College of Criminal Justice, Naif Arab University for Security Sciences, Riyadh 14812, Saudi Arabia

* Correspondence: nazrulh@upm.edu.my (M.N.H.A.); samer@nauss.edu.sa (S.A.A.); Tel.: +966-559822001 (S.A.A.)

**Abstract:** This review highlights the effect of carcinomas on the results of the examination of autosomal genetic traits for identification and paternity tests when carcinoid tissue is the only source and no other samples are available. In DNA typing or genetic fingerprinting, variable elements are isolated and identified within the base pair sequences that form the DNA. The person's probable identity can be determined by analysing nucleotide sequences in particular regions of DNA unique to everyone. Genetics plays an increasingly important role in the risk stratification and management of carcinoma patients. The available information from previous studies has indicated that in some incidents, including mass disasters and crimes such as terrorist incidents, biological evidence may not be available at the scene of the accident, except for some unknown human remains found in the form of undefined human tissues. If these tissues have cancerous tumours, it may affect the examination of the genetic traits derived from these samples, thereby resulting in a failure to identify the person. Pathology units, more often, verify the identity of the patients who were diagnosed with cancer in reference to their deceased tumorous relatives. Genetic fingerprinting (GF) is also used in paternity testing when the alleged parent disappeared or died and earlier was diagnosed and treated for cancer.

**Keywords:** DNA fingerprinting; genetic fingerprinting; autosomal trait screening; carcinomas

## 1. Introduction

In 1985, Jeffreys discovered DNA profiling, which is one of the most important and prominent developments in the security and criminal fields [1–3]. DNA profiling, also known as genetic fingerprinting (GF) or DNA typing, is the process by which a person expresses only one copy of a gene (either from the mother or the father) and suppresses the other copy. GF was used to identify people by comparing specific segments of DNA. It was also used to address parentage issues and filiation disputes. Several other fields, such as the medical, environmental, and agricultural sectors, also benefitted from the application of GF [1–3]. According to Jeffreys, who documented the initial development of multilocus DNA fingerprinting, these individual-specific DNA patterns could provide a powerful approach to personal identification and paternity testing. At the time, it was anticipated that these applications would take a while to develop and that significant legal problems would arise when DNA evidence moved from the research lab to the courtroom [1,2]. In April 1985, the first case involving a dispute over immigration to the UK was successfully resolved using DNA fingerprinting [1]. Shortly afterwards, a UK civil court accepted DNA evidence in a paternity case. The Enderby murder case in October 1986 marked the debut of DNA typing in criminal investigations. This investigation led to the first case of the release of a prime suspect who was later proved innocent by DNA evidence [1]. As early as 1987, DNA typing results were admitted as evidence in criminal courts in

the UK and the USA. In 1988, the UK Home Office and Foreign and Commonwealth Office approved the use of DNA fingerprinting to resolve immigration disputes involving disputed family relationships [1,3,4]. GF has developed and evolved by passing through several technical stages until it recently settled to employ a group of genetic sites, as these sites are characterised by high individual strength, are spread on all the chromosomes that make up the human genome and contain short-length tandem passages. Therefore, these sites are of great importance when examining decomposing forensic samples. The technique employed is called short tandem repeats (STRs). STR is the optimum technology used in examining and analysing genetic traits in fingerprinting laboratories, as it suits working conditions in the criminal and forensic fields [5,6].

Among the genetic sites, microsatellites (MS) are 1-6 base pairs of tandem repeats found within introns and are subject to insertion/deletion events [7]. They are polymorphic loci, as the number of their repeat units varies from one individual to another. When the variation in their number occurs in or near a gene, a change in the function of that gene could be produced. MS have acquired an important oncological interest as they represent the main sign of instability, which is recorded by the expansion or contraction of their sequences in tumour DNA as compared to the normal DNA from the same individual [8]. When multiple unstable genomic loci are found, the tumour is characterized by high-frequency microsatellite instability (MSI), while those displaying only one unstable genomic locus are referred to as low-frequency MSI. Microsatellite stability is considered when none of the analysed loci exhibits instability [9]. Microsatellite instability (MSI) is a molecular fingerprint arising because of defective mismatch repair genes (MLH1, MSH2, PMS1, PMS2, MSH6, or MSH3). Such defects can be inherited (i.e., germline insertion/deletion in the MLH1 gene) or sporadic (e.g., inactivation of MLH1 through hypermethylation of its promoter) [10]. MSI shows the accumulation of mutations in microsatellite DNA repeat sequences spread throughout the genome. It is considered a major biomarker for familial cancer risk assessment, cancer prognosis, and therapeutic choices. Mismatch repair genes are an evolutionarily conserved system preserving DNA homeostasis [11] by recognizing and repairing nucleotide mispairing or insertion/deletion generated during DNA replication, recombination, or damage [12]. Genetic and epigenetic inactivation of mismatch repair genes cause their defects, inducing genome cancer [13].

Cancer arises from a combination of the loss of tumour suppressor genes and the activation of oncogenes. Most oncogenes are activated by simple missense variants that are unlikely to affect the fingerprint. Loss of tumour suppressor genes, especially DNA repair genes, may increase the risk of multiple mutations throughout the genome. The loss of chromosomal material, in which the second copy of a tumour suppressor is often lost, can delete information important for fingerprinting [14]. Because many genetic mutations occur in the formation of the DNA strand in the human genome of cancer patients, it is possible that these mutations affect the composition of GF sites across different chromosomes, which leads to varying results of their examination for tumour samples of these patients. If undefined human tissues with cancerous tumours were obtained from sites of mass disasters or terrorist incidents, the reliability of the examination of the genetic traits from these tissues would be affected. This will result in a failure to identify the correct identity of the person [15].

In writing this review, answers to the following questions were sought:

1. What is GF, and what are the GF sites?
2. What is the role of GF in detecting cancerous tumours?
3. What are cancerous tumours, and what is the effect of carcinomas on GF?

## 2. GF Technique

The characteristics of all living organisms, including humans, are essentially determined by the information contained in the DNA they have inherited from their parents. The molecular structure of DNA can be thought of as a two-strand zipper, with nucleotides being the teeth of that zipper. Each tooth is represented by one of the four letters (A, C,

G or T), and the opposing teeth form one of two pairs, either A-T or G-C. The letters A, C, G and T stand for adenine, cytosine, guanine and thymine, the basic building blocks of DNA. The information contained in DNA is primarily determined by the sequence of letters along the zipper. For example, the sequence AAT stands for different information than the sequence TAA, although they use the same letters.

The process of GF was developed in 1985 by the geneticist Alec Jeffreys. He discovered that some sections of strands of DNA contained sequences of nucleotides repeated next to each other. He also discovered that these sequences are found in the same order of nitrogenous bases in all humans but differ from person to person regarding the number of times they are repeated [1]. They are fixed in their arrangement but differ in the number of repetitions. They were employed to differentiate among humans and determine individuals' identities. Since these repeated sequences are inherited and transmitted from parents to children, they can also be employed in the resolution of questions of paternity [1,2]. A repeated sequence of nitrogenous bases on a DNA strand is categorised according to its length—that is, the number of nitrogenous base pairs in the sequence—into one of three types (long, medium-length and short repetitive sequences), all of which are called satellites. Long repetitive sequences, or macrosatellites, contain hundreds to thousands of pairs of nitrogenous bases and are found in certain regions on the chromosomes of the human genome, specifically in the heterochromatic areas near the centromere and at the ends of telomeres; they are also found on the male Y chromosome. Medium-length repetitive sequences, also known as minisatellites, contain between 10 and 100 pairs of nitrogenous bases. They include the repetitive sequences found at a variable number of tandem repeat (VNTR) loci, which are present in some parts of the euchromatic regions on various chromosomes in the human genome. Short repetitive sequences, or microsatellites, contain 1 to 6 pairs of nitrogenous bases [7]. They include the repetitive sequences found at short tandem repeat (STR) loci, which are located in many parts of the euchromatic regions on all chromosomes in the human genome [16–18].

GF has evolved in two distinct stages. The first stage relied on the use of VNTR analysis to detect repeating sequences of medium length (minisatellites; 10 to 100 base pairs) and the subsequent use of the restriction fragment length polymorphism (RFLP) technique to analyse the genetic traits of the DNA in the genetic loci. The VNTR technique for GF analysis was exclusively used for approximately 10 years after its discovery [19]. In the mid-1990s, after the discovery of short repetitive sequences (microsatellites), the second stage of GF—STR analysis—emerged. This technique, also called simple sequence repeat (SSR) genotyping, examines the genetic traits of DNA through the analysis of genetic sites in which these short sequences are repeated. STR analysis is the latest GF technique, and it is currently the optimal technique for screening for genetic traits at DNA sites. This technique is easy and quick, and the analysis is performed in an automated manner; it is easy to amplify DNA segments using the polymerase chain reaction (PCR) technique because the repetitive sequences are very short (2 to 7 base pairs). Furthermore, this technique allows for clear and accurate results to be obtained on the same day, and the fact that these genetic sites are large and scattered on all chromosomes in the human genome, in addition to the presence of a high percentage of heterozygote genetic sites (approximately 70%), means that its power to differentiate between individuals is very high. Moreover, the STR technique is characterised by its ability to analyse several genetic sites simultaneously; it can also be used to analyse partial or very small quantities of repeats [19,20]. STR analysis is the preferred technique for the development and collection of latent print evidence in GF and forensic laboratories worldwide [21].

Genetic loci that contain short repetitive sequences are known as STR loci, STR markers or genomic fingerprinting loci. The human genome contains many STR markers scattered on all somatic and sex chromosomes, where they occur at a rate of 1 per 10,000 nitrogenous bases. The sites usually spread in the non-coding regions, and they are located either (1) outside the boundaries of genes, in the regions that separate genes on the DNA strand or (2) within the boundaries of genes, in what are called introns [22].

Genes consist of two parts: exons and introns. Exons, which are the functional parts of each gene in which the process of gene expression occurs, produce the proteins responsible for genetic traits. Introns are the non-functional parts of genes in which no gene expression occurs and no type of protein is produced. Consequently, GF sites are named according to their locations. A genetic site located within a particular gene is designated by the gene's name, while the genetic sites that lie outside the boundaries of known genes are named based on the chromosomal positions of the loci in the human genome. Each genetic locus on autosomal chromosomes contains two variants of the gene (called alleles) because there are two copies of autosomes in each cell (one inherited from each parent). The allele at a DNA site is the number of repeats of the sequence of nucleotides at that site. If the alleles are different, they appear as two different numbers, such as the genotype '9,7'; in this case, the genotype is heterozygous. If the alleles are identical (i.e., the same number of repetitions—for example, '8,8'), the genotype is homozygous. As for the fingerprinting sites on sex chromosomes, each genetic locus contains only one allele because each cell contains one copy of the sex chromosomes; a female carries the X allele at the locus on her chromosome, while a male carries the Y allele at the locus on his chromosome. The main 13 STR loci used for DNA fingerprinting analysis—the Combined DNA Index System (CODIS) core and extra STR loci—are located on the autosomal chromosomes of the human genome [23,24], in addition to the amelogenin on the sex chromosomes (Figure 1).

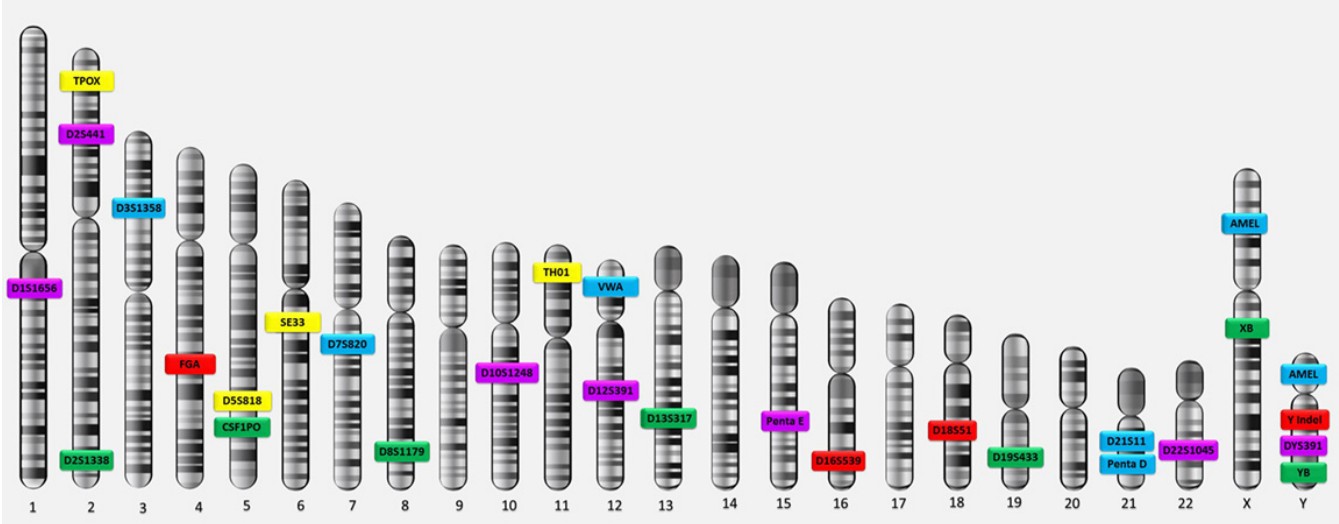

**Figure 1.** CODIS Core (TPOX, D3S1358, FGA, D5S818, CSF1PO, D7S820, D8S1179, TH01, VWA, D13S317, D16S539, D18S51, D21S11) and extra forensic STR loci on human chromosomes [25].

## 3. Medical Applications of GF in Carcinomas

A DNA fingerprint is the same for every cell, tissue, and organ in a person, unlike a traditional fingerprint, which is only found on the fingertips and can be altered by surgery. As a result, DNA fingerprinting is rapidly replacing other methods to recognise and distinguish unique people. An additional application of GF technology is the diagnosis of disorders and diseases in humans. Detecting disorders and diseases early allows the medical team to provide effective treatment. GF has been used in many medical applications, arguably the most important of which is the detection of cancerous tumours [26]. Cancer is a large group of more than 100 complex or composite diseases. In most cases, doctors usually perform a biopsy to confirm the presence of cancer. In a biopsy, a sample of the abnormal tissue is taken. A pathologist examines the tissue under a microscope and performs further tests on the cells of the sample. Although an elevated level of a circulating tumour marker can occasionally be helpful in diagnosing cancer and can indicate the presence of cancer, it is not sufficient on its own to make the diagnosis. Thus, it happens that the levels of certain tumour markers rise due to non-cancerous conditions [27]. These diseases differ

in their behaviours according to the types of cells from which they originated, but all types of cancer share two characteristics: (1) abnormalities in the processes of cell division and growth and (2) the ability of cells to spread (metastasise) and invade tissues other than those from which they originated. The main reason for the emergence of cancer is damage to DNA, specifically in the coding regions on chromosomes, which contain genes and are where gene expression occurs [28]. During this process, a defect or error occurs during DNA repair, which leads to a change in the sequence; when the DNA is doubled and cells divide, fixed genetic mutations occur because of that defect. Then, incorrect codes are copied and translated by messenger RNA, resulting in abnormal proteins; these may affect the functions of some genes (e.g., activating oncogenes or inhibiting tumour-suppressor genes that work to control and regulate cell growth). Therefore, cell growth and division continue abnormally, and apoptosis (programmed cell death) does not occur. When DNA damage occurs, signals are released during the cessation of division (resting period) to control the *p53* gene in the next stage, after which the cell cycle stops, allowing DNA repair; alternatively, the cell will die by apoptosis if the damage is not repaired. In some cells, the *p53* gene may not be active or stimulated because of cellular damage. This may lead to the failure of the cell cycle to stop at the control point, causing the cell to continue for the rest of its life cycle with errors or damage to the DNA without repair or apoptosis occurring. In this case, the cell accumulates the types of mutations or chromosomal changes that lead to cancer development [29].

Apoptosis is one of the vital processes necessary to maintain the positions of cells in tissues as well as their balance. During this programmed cell death process, a cell kills itself during growth via the shrinkage and decomposition of the contents of the nucleus and the disintegration of DNA into very small pieces, and the remains of the dead cell are consumed by the healthy neighbouring cells. Apoptosis can be inhibited by tumour-suppressing genes because of exposure to genetic mutations. This allows cells with damaged DNA to grow abnormally and become cancerous [30]. MS regions, or STR markers, are essential tools for mapping disease-causing genes in relation to both forensic investigations and population genetics studies (Figure 2).

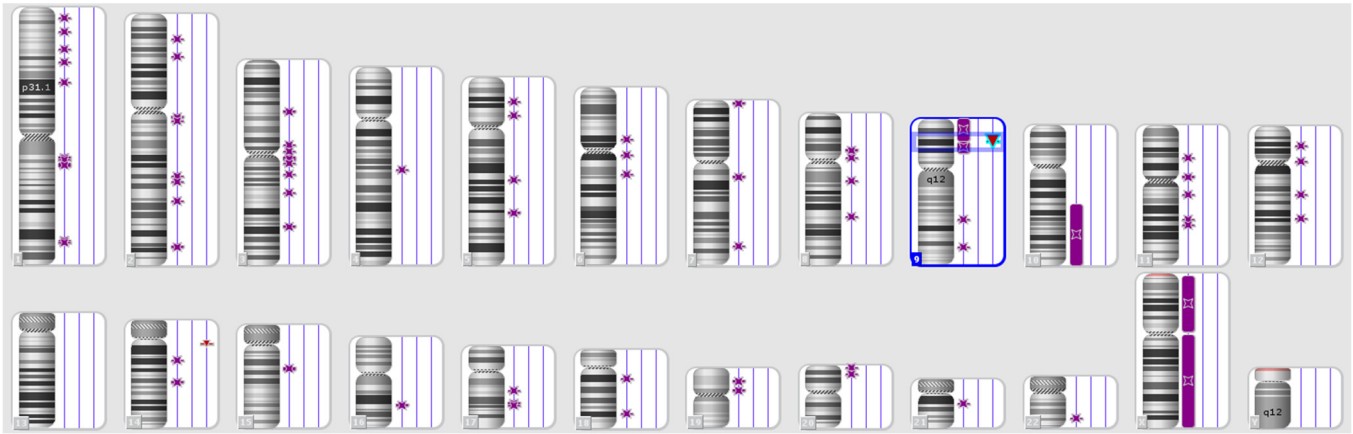

**Figure 2.** Karyotype of a patient with the loss of heterozygosity and normal karyotype according to the standard cytogenetic analysis [31]. Purple symbols refer to loss of heterozygosity.

They are also important for studying the genetic changes in tumours since tumour tissues often display some type of MSI as a result of mutations and somatic changes. Another type of genetic instability, loss of heterozygosity (LOH), is accompanied by allelic losses. Cancerous tumours are usually categorised in terms of instability, and somatic genetic changes in the microsatellite regions are divided into three subgroups. First are tumours with a very high level of microsatellite instability (MSI-H), in which the genetic loci have a mutation rate of more than 30%. Second, are tumours with a low level of microsatellite instability (MSI-L). The third group comprises tumours with microsatellite

stability (MSS). LOH is a common method of detecting genetic deletions in many tumours. LOH is indicated by severe allelic defects at one or more genetic loci in a single-source DNA sample; one allele is deleted, and the other allele remains at the heterozygous locus and appears as homozygous. In this case, LOH influences human identification when an archived clinical sample from a tissue biopsy is used as a reference sample for the identification of a person in mass disasters (Figure 3). In many cases, the LOH may be expressed in terms of a decrease in the length or height of allelic peaks that appear in the result of STR analysis and genetic trait examination [32,33].

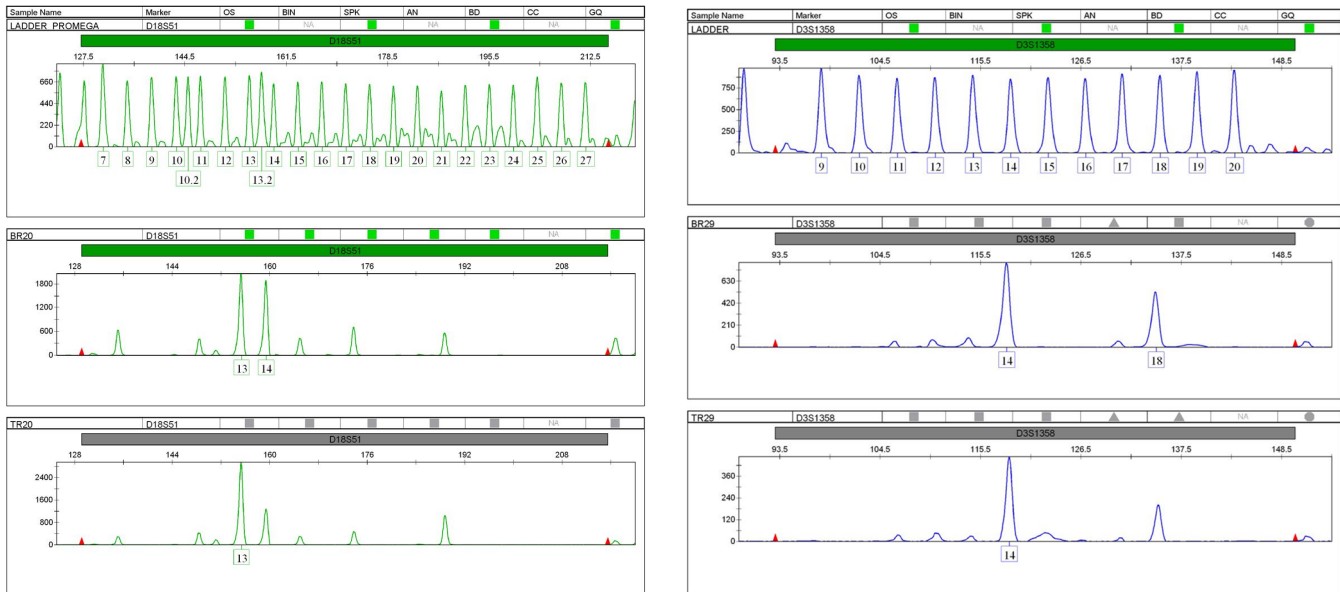

**Figure 3.** Loss of heterozygosity in STR profiles (D18S51 and D3S1358) of the patient's breast tumour compared to the STR profile of her normal tissue [34].

## 4. Effects of Carcinomas on Genetic Fingerprints

Several studies have linked the incidence of various cancerous tumours (and the differences resulting from these tumours) to the alleles of the genetic fingerprints of affected individuals. However, other studies have described differences in patients' genetic fingerprint analyses, indicating that cancerous tissue samples cannot reliably identify individuals or resolve questions about paternity. A study by Al-Harthi [34] was conducted to determine the effects of breast cancer on the results of autosomal STR marker profiling on thirty-one cancer tissue samples from breast cancer patients in Riyadh city. The study followed the practical experimental methodology using the available laboratory techniques in the field of STR profiling of DNA isolated from cancer tissues. The study found that breast cancer can lead to genetic mutations in up to 54.84% of cases. The analysis detected genetic mutations in 18 of 22 autosomal STR loci (75%). Genetic mutations occur in metastatic breast cancer tumours that can migrate to other sites in the body. Loss of heterozygosity (LOH) is the main genetic mutation type in breast cancer tumours. The study did not recommend relying on the results of STR genotyping of breast cancer tissue samples, either for human identification or paternity testing.

Another study by Much et al. [35] in the United States aimed to detect changes in the results of identification tests of tissues with MSI. Among 21 carcinoma samples, 11 (nine intestinal adenomas and two endometrial carcinomas) were MSI unstable, while 10 (colon and rectal adenocarcinomas) were MSI stable. The study reported that MSI may significantly alter the wild-type allelic polymorphism, leading to potential interpretation errors in STR genotyping. In 10 of the 11 MSI-unstable samples, new alleles appeared at several genetic loci (from 5 to 12 STR loci in each case, often with three or more allelic peaks), and in six of the 11 cases, LOH was observed. Allelic traits of the genotype were seen in

seven out of the 10 MSI-stable samples, and LOH occurred in eight out of the 10. The careful examination of the STR allelic pattern, a high index of suspicion and follow-up MSI testing are crucial to avoid erroneous conclusions and subsequent clinical and legal consequences.

A study by Pelotti et al. [36] was conducted among 56 different gastrointestinal cancer samples (stomach and intestinal carcinomas) with the aims of providing more data, assessing the incidence of allelic changes in 15 genetic loci (15 STR loci) and determining whether the changes were the result of MSI or LOH. The study reported that 66% of the cancerous tissues possessed allelic alterations of the microsatellites analysed, with a high incidence of MSI-L when compared with the corresponding normal tissues. The most frequently altered loci were D18S51, VWA, and FGA. From a forensic perspective, great care must be taken when evaluating DNA typing results obtained from cancerous tissue samples.

Additionally, Ceccardi et al. [37] evaluated the genetic profiling screening of STRs used for forensic identification among 68 human tumour tissues, including 48 gastrointestinal cancer samples, 13 urogenital cancer samples and seven oral cancer samples. The study reported that a total of 37 cancerous tissues (54.4%) showed allelic alterations when compared with the corresponding normal tissues. These included 29 (78.4%) gastrointestinal tumours, 4 (10.8%) urogenital tumours, and 4 (10.8%) oral tumours. The loci most frequently affected by allelic alterations were VWA, FGA, and D18S51. These results suggest that great care should be taken in evaluating DNA typing results obtained from clinical cancerous specimens when no other reference samples containing normal tissue are available.

A study by Alharbi et al. [38] investigated the impact of leukaemia on the detection and stability of short tandem repeat (STR) markers. The study involved analysing DNA samples from 15 individuals with chronic myeloid leukaemia (CML) and 15 healthy controls. The researchers found that individuals with leukaemia had a higher number of alleles at the tyrosine hydroxylase 1 (TH01) marker compared to the control samples. The results suggest that STR markers could be useful in genetic studies of leukaemia cases. STR markers have been widely used in various applications such as forensic DNA analysis, paternity testing, and genetic disease detection. They have high discrimination power and can provide valuable information in identifying and monitoring diseases, including different types of cancer. The study highlights the potential of using STR markers in studying leukaemia and understanding its genetic characteristics.

In addition, Al-Qahtani et al. [39] reported on the use of 16 autosomal short tandem repeat (STR) loci in forensic investigations using samples from colorectal cancer (CRC) patients. The study aimed to evaluate the genetic data obtained from CRC tissues and compare it with the adjacent non-cancerous tissues. The results showed that there were no significant differences in the genetic information between CRC and non-cancerous tissues, but there were genetic deviations in certain loci. These deviations could potentially affect data interpretation, especially in regard to false homozygosity/heterozygosity and misinterpretation of DNA profiles. The study also demonstrated the potential application of CRC tissues as a DNA source for forensic investigations. However, it is important to consider the microsatellite instability (MSI) and loss of heterozygosity (LOH) effects present in cancer tissues, which may complicate data interpretation. Overall, the findings contribute to the field of forensic science, particularly in using genetic markers for CRC diagnosis.

A study explored the use of a forensic short tandem repeat (STR) kit to detect somatic hypermutational tumours in gastrointestinal (GI) cancers. The researchers tested 250 GI tumour samples using two commercial kits to determine microsatellite instability (MSI) and STR alterations. They found that 62.4% of patients exhibited STR alterations, including 100% of MSI-H cases and 60% of MSI-L/MSS cases. The researchers also identified three types of forensic STR alterations: allelic loss, Aadd, and Anew. The allelic loss was the most common alteration observed in GI tumours. The study suggests that the widely used forensic STR kit may have potential usage in identifying hypermutational status in GI cancers. This method could help in the selection of patients who can benefit from immunotherapeutic agents [40].

Another study examined the genetic alterations and DNA damages caused by formalin-fixed tumour tissue in forensic identification. Tumour tissue samples from 25 patients who had undergone surgery for neoplasia were analysed. The DNA profiles of both fresh and formalin-fixed tumour tissue were compared to those of fresh and formalin-fixed normal tissue from the same patient. The results showed that only a quarter of the samples had the same genotypes in all tissue specimens, while the rest had at least one altered locus. The most common genetic alterations observed were partial loss of heterozygosity (pLOH), complete loss of heterozygosity (LOH), additional alleles, and substitution of new alleles. The frequencies of pLOH and LOH were higher than allelic addition or substitution. The study also identified specific genetic markers that were frequently altered in tumour tissue, including D5S818 and D18S51. These findings highlight the importance of considering genetic alterations caused by formalin fixation when using tumour tissue for forensic identification [41].

A study by Chen et al. [42] investigated the status of short tandem repeats (STRs) and microsatellite instability (MSI) in different tumour types. The study analysed 407 paired DNAs from eight tumour types: breast cancer, hepatocellular carcinoma, pancreatic cancer, colorectal cancer, gastric cancer, lung cancer, oesophageal cancer, and renal cell cancer. The results showed that the frequency of STR changes varied between the different tumours, with the highest frequency observed in oesophageal cancer and the lowest in pancreatic cancer. Interestingly, none of the patients in the study had MSI-low or MSI-high, except for patients with gastrointestinal cancer. The study also predicted potential thresholds for hypermutability in different tumour types based on published objective response rates. The results suggest that STRs may be an alternative marker for assessing hypermutability in different tumour types, but further clinical studies are needed to validate these observations.

Another study looked at the clonal evolution of tumour cells and the potential implications for therapy selection. The researchers conducted an in vitro study using cultured tumour cell lines (Jurkat) and non-tumour cell lines (WIL2-S). They analysed the subclones of these cell lines using Short Tandem Repeats (STR) profiling to identify genetic aberrations. They found that new aberrations occurred more frequently in tumour cells than in non-tumour cells. They also found a significant correlation between the accumulation of aberrations and the rate of cell growth. These results suggest that tumour cells have a higher potential for genetic instability and clonal evolution. The researchers suggest that this approach could be used to analyse primary tumour cell cultures from patients to aid in therapy selection [43].

In addition, a study investigated the reliability of using malignant tissue samples for forensic investigations by analysing Short Tandem Repeat (STR) loci in surgically removed samples of papillary thyroid cancer (PTC). The study found four types of changes in STR loci between normal and tumour tissue: partial loss of heterozygosity (pLOH), complete loss of heterozygosity, additional alleles, and new alleles not found in normal tissue. These changes were observed in 20 of the 23 STRs tested, with D2S1338 being the most affected locus and pLOH being the most common change. The study suggests that caution should be exercised when interpreting DNA typing results from malignant tissue, especially when reference samples from normal tissue are not available. Patients aged 40–59 years had the highest frequency of STR variation. The results contribute to the understanding of the genetic mechanisms underlying PTC and highlight the challenges associated with the use of tumour tissue in forensic casework [44].

In addition, a review study reported that forensic short tandem repeat (STR) markers, commonly used for DNA profiling, could provide information beyond mere identification. The authors conducted a comprehensive search and identified 57 studies that reported an association between a STR-inclusive gene and a phenotype. These studies identified 50 unique traits associated with the 24 markers included in the analysis. The STR marker TH01 had the largest number of associations, including five studies that linked it to schizophrenia. However, none of the associations found were independently causative or predictive of disease. While forensic STRs have traditionally been considered non-coding and unrelated

to phenotype, there is increasing evidence that STRs in non-coding regions may influence gene expression and contribute to specific traits. Further research is needed to explore the potential functional role of forensic STRs [24].

The cancerous tissue is usually heterogeneous, with some having a high proportion of normal cells, and it is usually removed with normal tissues bordering it that can be used easily for genetic fingerprinting. If no normal material is available, areas with very low neoplastic content can be identified and, therefore, still may contain at high levels any markers used in fingerprinting; as such, low allele levels of a marker such as an SNP can be discarded in most cases as being due to low level of contamination with tumours. Even in biopsy samples containing high neoplastic content, available assessment of rare polymorphisms may still be possible. However, genetic instability and degradation in archived histology samples from cancerous tumours could affect the reliability of STR typing and its potential for human identification [45]. Fingerprinting of formalin-fixed-paraffin-embedded (FFPE) tissue DNA from deceased relatives is an accurate and informative tool in the clinical management of families where an inherited cause of cancer is suspected. GF of DNA obtained from germline is recommended because if a pathogenic variant is present in the family, it is these individuals in whom it is most likely to be detected, particularly if their tumorous relatives were deceased [46]. GF counselling, risk assessments, screening recommendations, and fingerprinting in families where an inherited cause of cancer is suspected are clinically applied with reliable diagnosis [47].

## 5. Conclusions

The above review has documented that GF became a precious method for solving crimes and protecting human rights, particularly by distinguishing people from one another, especially discovering the criminals and proving paternity in family conflicts or identifying the dead and the missing. The STR technique is used to employ a group of genetic sites on the DNA, as these sites are characterized by the fact that they spread on all the chromosomes that make up the human genome and that they are of high individual strength, as well as contain short-length tandem sections. This made it of great importance in cases of examining decomposing forensic samples. However, there is a possibility that the mutations in carcinomas affect the formation of GF sites spread on different chromosomes, which leads to different results of the genetic fingerprint examination for carcinoma samples. This may affect the examination of the genetic traits derived from these samples, which may result in a person not being identified correctly.

**Author Contributions:** Conceptualization, H.A. and M.N.H.A.; methodology, H.A.; investigation, M.N.H.A.; writing—original draft preparation, H.A.; writing—review and editing, M.N.H.A., S.A.B. and S.A.A.; supervision, M.N.H.A., S.A.B. and S.A.A.; project administration, M.N.H.A. All authors have read and agreed to the published version of the manuscript.

**Funding:** This research received no external funding.

**Acknowledgments:** This review would not have been possible without the help of those who contributed.

**Conflicts of Interest:** The authors declare no conflict of interest.

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
