# Peer review of "Effect of Carcinomas on Autosomal Trait Screening: A Review Article"

_cimb, doi:10.3390/cimb45090460_

Round 1
Reviewer 1 Report
Comments and Suggestions for Authors
The authors reviewed the impact of assessing tumour material as the only identified source for identifying a person from autopsy or other situations. They rightly point out that this can affect genotyping and the accuracy of using certain genomic fingerprinting techniques but make no attempt to discuss how to get around this such as dissecting normal material from tumour samples or using tissue with low neoplastic content. They also conflate a number of issues such as imprinting with fingerprinting and I’m not really sure why they even mention imprinting.
Specific points
1. Abstract ‘If these tissues are infected with cancerous tumours, it may affect the examination of the genetic traits derived from these samples, thereby resulting in a failure to identify the person.’ Cancer is not an ‘infection’ please reword
2. ‘Genetic finger-printing also known as genetic fingerprinting’ -This is the same thing? The Hyphen does not make it different
3. ‘Genomic fingerprinting does not change the DNA sequence itself, unlike genomic mutations that can affect how inherited genes are expressed. Instead, gene expression is silenced by the epigenetic insertion of chemical marks into the DNA during egg or sperm production’ This is NOT the correct terminology. I think you mean here ‘imprinting’ not ‘fingerprinting’!
4. ‘The genetic imprint has developed and evolved by passing through several technical stages until it recently settled to employ a group of genetic sites on the DNA tape, as these sites are characterised by high individual strength, are spread on all the chromosomes that make up the human genome and contain short-length tandem passages.’ -You keep conflating imprinting with fingerprinting. You are not describing ‘imprinting’ here
5. ‘If undefined human tissues obtained from sites of mass disasters or terrorist incidents are (infected) with cancerous tumours, the reliability of the examination of the genetic traits from these tissues will be affected.’ -reword please
6. ‘The main cause of cancerous tumours is damage to the DNA as a result of genetic mutations. This leads to the stimulation of oncogenes that activate the abnormal and uncontrolled growth and division of normal cells.’ This is far too simplistic. Cancers occur because of combinations of loss of tumour suppressor genes and activation of oncogenes. Most oncogenes are activated by simple missense variants that are unlikely to affect fingerprinting. Loss of tumour supressors and particularly DNA repair genes may give risk to mutilple mutation throughout the genome. Loss of chromosome material which often loses the second copy of a tumour suppressor may delete information important for fingerprinting
7. ‘For example, the sequence ACGCT stands for different information than the sequence AGTCC, just as the word "POST" has a different meaning than "STOP" or "POTS"," This is not a good example. DNA readouts are in base triplets not 5 bp.
8. The main issue with this review is the failure to even mention that cancerous tissue is usually heterogenous with some having a high proportion of normal cells. Most cancers that are removed have normal tissue bordering the cancer that can be used easily for genet fingerprinting. Even if no normal material is available areas with very low neoplastic content can be identified and therefore will still contain at high levels any markers used in fingerprinting. As such low allele levels of a marker such as a SNP can be discarded in the vast majority of cases as being due to low level contamination with the tumour. Even in samples such as if only a biopsy containing high neoplastic content. If this is all that is available assessment of rare polymorphisms only may still be possible
9. Use of tissue from removal of tumours on deceased relatives can be sued to identify traits with especially use of the non-tumorous element https://pubmed.ncbi.nlm.nih.gov/33654310/
Comments on the Quality of English LanguageNot too bad
Reviewer 2 Report
Comments and Suggestions for Authors
The authors reviewed the application of genetic fingerprinting in the identification of a person when the only available tissue for analysis is cancer tissue. They discuss the role of microsatellite instability in cancers so that tumor tissue is not very good for forensic analysis. The authors specify that the reasons for using such tissue for analyses are mass catastrophes and terrorist incidences when human remains which were found in place of catastrophe are infected with cancerous tumors and fragments of normal tissue are not available. I have doubts about this aim of reviewing the topic. I assume that such situations happen very rarely. However, that are other reasons that the cancer tissue is used in genetic fingerprinting. Patients who were diagnosed with cancer sometimes need verification whether they got the proper diagnosis because of the possible mix of samples. Since paraffin blocks are stored at pathology units they can be used to verify the identity of the patients and I assume that such situations happen more often than mass catastrophes. The other reason of using such samples is paternity test when the alleged parent disappeared or died and earlier was diagnosed and treated for cancer.
The authors provided in the introduction the basic information about molecular biology, genetic fingerprinting, genetics of cancer and microsatellite instability. For the reader who doesn’t know very much about these topics this information in introduction is most likely incomprehensible. I would suggest that the authors should focus on genetic fingerprinting and microsatellite instability in cancer as an introduction to the topic they want to review. The Fig. 1 is very good and authors should also add the cancer karyotype picture so the readers will see the problem with fingerprinting in cancer tissue very well.
Reviewer 3 Report
Comments and Suggestions for Authors
Review for the manuscript “Effect of Carcinomas on Autosomal Trait Screening: A Review Article”
The authors tried to show the role of the genetic fingerprinting method for solving crimes, for distinguish people from one another, also for proving paternity in family conflicts.
They presented a lot of information regarding genetic fingerprinting technique, also medical applications of this method in carcinomas and effects of carcinomas on genetic fingerprints. They conclude that the presence of different mutations carcinomas affect the formation of genetic fingerprinting sites which may result in a person not being identified correctly.
Comments
The study is very interesting and the results are important for clinicians.
The presentation was clear. The writing style is also clear.
The references are adequate.
Please provide information how did you collect the results for this article? Which type of scientific databases are you searched?
Round 2
Reviewer 1 Report
Comments and Suggestions for Authors
The authors have responded well to reviewer comments and the manuscript is much improved
Comments on the Quality of English LanguageFine
Reviewer 2 Report
Comments and Suggestions for Authors
The manuscript has improved. The authors added my suggestions to the introduction about the reasons of using tumour tissues for genetic fingerprinting to identify the person. They also added the fragment describing the paraffin blocks containing very often the segment of tumour and normal tissue so in rare situations when only tumour tissue is available in paraffin block, it must be used for genetic fingerprinting. They added also more information about microsatellite instability.